# Effect of Different Clarification Treatments on the Volatile Composition and Aromatic Attributes of ‘Italian Riesling’ Icewine

**DOI:** 10.3390/molecules25112657

**Published:** 2020-06-08

**Authors:** Teng-Zhen Ma, Peng-Fei Gong, Rong-Rong Lu, Bo Zhang, Antonio Morata, Shun-Yu Han

**Affiliations:** 1Gansu Key Laboratory of Viticulture and Enology, College of Food Science and Engineering, Gansu Agricultural University, Lanzhou 730070, China; matz@gsau.edu.cn (T.-Z.M.); qpxhgpf@163.com (P.-F.G.); lurongrong0416@163.com (R.-R.L.); zhangbo@gsau.edu.cn (B.Z.); 2Food Technology Department, Technical College of Agricultural Engineers, Technical University of Madrid, Avenida Complutense S/N, 28040 Madrid, Spain; antonio.morata@upm.es

**Keywords:** fining, filtration, icewine, volatile compounds, aroma series, sensory analysis

## Abstract

The aim of this study was to evaluate the influence of clarification treatments on volatile composition and aromatic attributes of wine samples. ‘Italian Riesling’ icewines from the Hexi Corridor Region of China were clarified by fining agents (bentonite (BT) and soybean protein (SP)), membrane filtration (MF), and centrifugation (CF) methods. The clarity, physicochemical indexes, volatile components, and aromatic attributes of treated wines were investigated. Both the fining agents and mechanical clarification treatments increased the transmittance and decreased the color intensity of icewine samples. Bentonite fining significantly influenced the total sugar content, total acidity and volatile acidity. Total acidity decreased 2–3.5% and volatile acidity 2–12%. MF showed the greatest influence on total phenol content, decreasing the initial content by 12%, while other treatments by less than 8%. Volatile analysis indicated that both the categories and contents of volatile compounds of wine samples decreased. MF treatment showed the most significant influence, while SP fining showed much lower impact. Odor activity values indicated the compound with the highest odor activity in Italian Riesling icewines was β-damascenone. For this compound, BT and SP did not show significant differences, however, in MF and CF it decreased by 20% and 63%, respectively. Furthermore, with high impact on aroma were: ethyl hexanoate which reduced by 20–80% especially in MF; rose oxide which extremely reduced in MF and undetected in BT, SP, and CF; isoamyl acetate which reduced by 3–33% and linalool decreased by 10–20% and undetected for BT. Principle component analysis indicated that icewine clarified by different methods could be distinguished and positively correlated with odor-active compounds. Floral and fruity were the dominant aroma series in icewine samples followed by fatty, earthy, spicy, vegetative and pungent flavor. The total odor active value of these series significantly (*p* < 0.5) decreased in different clarification treatments. Sensory evaluation showed similar results, but the SP and CF wine samples achieved better sensory quality. This study provides information that could help to optimize the clarification of ice wines.

## 1. Introduction

Clarity is an essential quality attribute of wine and is recognized and valued by consumers together with the taste and aroma properties. Nowadays, wine stabilization and limpidity can be enhanced by centrifugation, filtration, fining or other clarification technologies [1]. The aim of this process is to decrease turbidity by removing suspended and colloidal particles in wine, meanwhile, macromolecules such as unstable proteins, which can later be denatured or aggregated and cause stability problems can also be removed [2,3].

The simplest and most economical way of clarification is natural settling or spontaneous sedimentation, where particles causing the turbidity settles to the bottom of wine by gravity. However, this method is time consuming and there are some particles with poor settling characteristics that cannot settle [3]. The aim of centrifugation is to accelerate the settling process by rotating it very fast around an axis. It is a rapid method for removing sediments and obtaining clean, stable and ready-to-drink wines, however, this system is restricted by the volume capacity or the enormous investment cost [3]. Filtration could eliminate solids or particles by passing wines through a filter medium, however, this separation technique was sometimes restricted by the clogging of filter surfaces, the throughput, the efficiency, the cost, and the practicability [3,4]. Clarifying agent addition (fining) is a process used to modulate and protect the organoleptic properties of the wines and to ensure their physicochemical stability by preventing the formation of hazes and deposits [5,6,7]. Unlike filtration and sedimentation that remove mainly particulates, fining could also remove some soluble substances including aroma components, coloring matter, and polymerized tannins in wines [8].

Bentonite, polyvinylpolypyrrolidone (PVPP), and protein-based fining agents are widely used in wine clarification. Among them, bentonite was the most efficient fining agent to obtain wine protein stability, however, its non-selectiveness may reduce both wine quality and quantity [9]. Horvat et al. found that bentonite added during fermentation positively affected wine quality by enhancing the preservation of key fermentation volatiles in relation to the control and exhibited positive sensory effects [9]. Other fining agents also influence wine aroma compounds, as Gil et al. found that thiol compounds in rose wine fining were PVPP dose-dependent, and a possible explanation was that PVPP would adsorb glutathione-S-conjugates aroma precursors, thus reducing the aroma content of the finished wine [7]. Proteins used for fining could come from animal origin (including egg albumin, casein, serum albumin, gelatin and isinglass) or plant origin (obtained from wheat, rice, pea, lupin and maize). However, the issues involved with animal diseases and their possible transmission to human beings have led the restriction on the use of animal products in wine fining, which increased the viable possibility use of plant-derived proteins [8,10,11]. As liquid chromatography-mass spectrometry analysis has detected the residual egg and wheat protein in wines, the use of these proteins was restricted by their potential allergenic features [12,13]. As a consequence, the plant-based products with lower allergenic potential such as soybean protein, were in response to winemakers’ interest [8,10,12]. Recently, vegetable protein fining was studied by numerous researchers in wine clarification. Granato et al. found that lentil proteins and gluten fining showed a significant influence on the total content of fermentative aroma compounds (esters and alcohols), while soybean protein showed less impact on wine aroma components [8]. Although the influence of clarification on wine volatile compounds have been identified, further study should be considered because only a limited number of volatile fractions can be perceived and contribute to wine aroma.

Icewine is a kind of sweet wine produced by naturally frozen grapes [14]. The contents of sugar, acids, and aroma compounds are concentrated in the ice-grape must which results in icewine having a rich, balanced, and intense flavor profile [15]. To ensure that top quality in icewines, aroma profiling tends to be an important priority [16]. Quality and odor profile of icewines were affected by many aspects from grape growing, winemaking, and aging technology. Among them, clarification tends to be an important technology because the high soluble solids levels of icewines make the clarification more difficult than dry wines, while a strength clarification treatment could significantly impact the final flavor and sensory quality of bottled wines. In this study, the effect of different clarification methods: membrane filtration (MF), centrifugation (CF), bentonite (BT) and soybean protein (SP) on clarity, physicochemical indexes, aroma quality and sensory characteristics of ‘Italian Riesling’ icewine were investigated, thus to provide useful insight and technical support in icewine clarification.

## 2. Results and Discussion

### 2.1. Chemical Composition and Transmittance of Wines

The first requirement for wine fining is to decrease turbidity or increase transmittance, thus, to get a crystal clear and translucent wine with good stability, whiles minimizing the influence on other parameters of the wine [2]. As can be seen from Figure 1A, both fining agents and mechanical clarification caused a significant increase in wine transmittance. All the treated wine samples showed a transmittance higher than 90%. Among them, MF treatment showed higher efficiency in wine clarification, which exhibited a significantly higher transmittance value (97.98%) compared to the other treatments. On the contrary, color intensity (CI) showed a significant decrease in treated wines (Figure 1B). The MF treatment showed the lowest CI value with a 35.44% decrease while SP fining only decreased by 9.54%. In conclusion, the effectiveness of the clarification would depend on the different treatments, both mechanical clarification and fining agent could increase wine transmittance, but decrease the wine color, among them, SP fining could clarify the wine and minimize the decrease in wine color intensity. This is in agreement with what other studies have found [8].

The chemical composition of the wines after clarification are shown in Table 1. Most parameters of the wine showed a decrease after clarification treatments, but the major differences revealed at total acidity, volatile acidity, and total phenolic content. BT fining showed the most significant decrease in total sugar and total acidity content, which may change the flavor, sweetness, and acidity of the wine, and was confirmed in sensory analysis. Total acidity was decreased by 2–3.5% and volatile acidity by 2–12%. BT and CF treatments decreased the protein content while wine treated by SP fining showed an increase. However, the change was not significant. Other researchers have reported that plant protein fining did not influence wine protein stability [8], and it could be settled together with wine suspended solids during the fining process [16]. Total phenol content showed a significant decrease in BT, SP, and MF treatments, which may delay or reduce the browning of the wine as Cosme and Laborde et al. reported [17,18]. MF showed the greatest influence on total phenol content decreasing the initial content by 12%, while other treatments by less than 8%.

### 2.2. Volatile Compounds Analysis

It was reported that over 1000 volatile compounds were identified in wine. The chemical classes and character of volatile compounds, the contents, and sensory impacts influenced wine flavor complexity and was pursued by the consumers [18,19,20]. The interaction between clarification treatments on aroma compounds depended on chemical features of the target compounds, the membrane materials, the physical-chemical characteristics of the fining agent, and the possible interactions between volatiles and other macromolecules previously linked to the fining agent [1,8,21,22]. In order to evaluate the impact of the fining agent or mechanical clarification on wine aroma components, solid-phase microextraction (SPME) coupled with a gas-chromatography-mass spectrometer (GC/MS) were used for the identification of volatile compounds.

As can be seen from Table 2, 57 volatile compounds were identified, including 20 esters, 14 alcohols, 9 fatty acids, 9 terpenes and norisoprenoids, 3 carbonyls, and 2 volatile phenol compounds. Compared to the control wine, the categories, and contents of volatile compounds changed significantly. However, SP fining showed lower influence compared to the other treatments, which has been found to be in agreement with Granato [8,12].

#### 2.2.1. Esters

Ester is one of the most important volatile species in wine aroma. It could be processed through the chemical interactions between alcohols and acids or released by yeast strains during alcohol fermentation [19,20,23]. The esters could mainly be categorized as acetate esters, ethyl esters, and other esters, which could contribute to wine fruity aroma [20,23]. A total of 20 esters were found in the experiment wine samples (Table 2) and among them, the control wine showed the highest level in both the quantities and contents of volatile compounds. MF treatment showed the most significant influence in ester compounds, however, 9 of them were not detected and the total contents reduced by 59.87%, followed by BT fining with a 47.74% decrease, while SP fining wine showed only a 16.97% decrease in total ester content and without the absence in quantities. This is in agreement with what Vincenzi et al. found. Bentonite treatment resulted in significant removal of ethyl esters and fatty acids [24]. Ethyl hexanoate, ethyl octanoate, ethyl decanoate and ethyl dodecanoate were the dominant ethyl esters present in wine in appreciable concentrations, while ethyl acetate and isoamyl acetate were the most abundant acetate esters. Most of the esters followed the same decrease tendency in treated wines. MF wine showed the lowest concentration, whereas the contents of ethyl butyrate, diethyl succinate and diethyl malate were lower in BT fining wine.

#### 2.2.2. Alcohols

Alcohol is another important volatile compound in wine which is mainly produced through yeast metabolism during wine fermentation and could contribute pungent scents to wine aroma. The aromatic contribution of alcohol depends on its concentration in wines. They could provide a pleasant aroma when their concentration bellows at 300 mg/L, on the contrary, a pungent aroma would be perceived [23,25]. A total of 14 alcohols were found in the experimental wine samples (Table 2). There was no significant difference between control wine and SP fining wine samples in total alcohol content, however, other clarification treatments showed a significant decrease, especially the wine treated by the MF. There was no obvious influence between BT and CF treated wines. Pentanol, 1-hexanol, and phenethyl alcohol showed relatively higher concentration among alcohols, however, unlike pentanol, the concentration of 1-hexanol and phenethyl alcohol only showed a significant decrease in MF treated wines.

#### 2.2.3. Acids

Acids were formed enzymatically during fermentation through yeast metabolism and were reported to contribute fruity, cheese, fatty, and rancid notes to wine aroma complexity at low concentrations. However, a negative flavor would occur if the concentration is too high [23,25]. A total of 9 acids were detected (Table 2). Among them, acetic acid, hexanoic acid, octanoic acid, and decanoic acid were recognized as the dominant acids. Compared to the control wine, clarification treatments reduced the acid concentration significantly, especially wine treated with the MF method, reduced by 38.81%, while CF wine only showed a 4.65% decrease. However, there was no obvious difference in the concentration of acid compounds between the control and the CF wines except for octanoic acid.

#### 2.2.4. Terpenes and C13 Nor-Isoprenoids

Terpenes and C13 nor-isoprenoids which are derived mainly from the grapes could be from the free aromas or the glycosidic aroma precursors and are known to be varietal aroma compounds [23,26]. As can be seen from Table 2, a total of 9 compounds were identified in this experiment. Compared to the control wine, clarification treatments significantly influenced the total concentration of terpenes and C13 nor-isoprenoids compounds, and among them, BT fining showed the greatest influence with a 32.36% decrease. This finding was in contrast with Vincenzi et al. [24], who found bentonite showed a low effect on the loss of terpenes, but the doses of bentonite were much lower than this experiment. On the contrary, wines treated by SP and CF showed a lower decrease with 10.34% and 13.36%, respectively. linalool, α-terpineol, and citronellol were the dominant terpenes identified. The concentration of linalool was significantly influenced by CF and BT treatments, while the concentration of α-terpineol and citronellol only showed significant decrease in MF wine. However, there was no observed difference in other treatments. β-damascenone and geranyl acetone belonged to C13 nor-isoprenoids, surprisingly, the influence of clarification treatments on these compounds was in controversy. Fining agents, both BT and SP treatment influenced geranyl acetone significantly, while mechanical clarification CF and MF treatments showed a significant impact on β-damascenone concentration.

#### 2.2.5. Carbonyls and Phenols

Three carbonyl compounds (two aldehydes and 1 ketone) were detected in wine samples (Table 2). Although the concentration of carbonyl compounds was relatively low in wine, they could contribute to the overall aroma through a synergetic effect [20,23]. 6-Methyl-5-hepten-2-one was not detected in wine samples treated with both fining agents, while oct-2-enal and furfural were not identified in MF wine samples. The total content of carbonyl compounds significantly decreased in MF wine while BT and SP wines showed much less impact. Volatile phenols, which formed by the action of hydroxycinnamate carboxylase, originated by the hydrolysis of precursors known as phenolic acids or hydroxycinnamic acids in grapes [27,28]. Clarification treatments decreased the concentration of volatile phenols significantly, especially the MF treatment, while CF treatment showed the least influence.

As has been discussed, MF treatment showed the most significant decrease in the total content of different volatile compounds, except for terpenes, which showed the maximum decrease in BT fining wine samples. Vincenzi et al. [24] suggested that the mechanism of aroma losses after bentonite fining may be due to the direct adsorption of the clay. In this experiment, we also confirmed this. On the contrary, SP fining showed the least influence on esters, alcohols, and terpene compounds, while CF and BT wines showed the least influence on fatty acids, phenol, and carbonyl compounds respectively. There was no significant difference between SP and CF wine in total terpenes. Additionally, the total carbonyls in BT and SP wines also showed no observed difference. The influence of membrane filtration on wine aroma quality could be influenced by the types of filtration, different filter media, filtration parameters and the membrane itself [3,4]. Recently, researchers have found that cross-flow filtration, a relatively new technique, produced the highest decreases in both total phenol index (TPI) and color intensity (CI) values and removed highly polymerized phenols that are related to astringency in red wines, therefore, could be used in the same way as fining [1,29]. In this study, MF treatment showed the most significant difference in volatile compounds of ‘Italian Riesling’ Icewine, but the mechanism still remains to be further studied.

### 2.3. Odor Activity Values

The odor activity value (OAV) was frequently used to indicate the contribution of the volatile compounds to the overall aroma in the wine [25,26]. Generally, a volatile with OAV (>1) was suggested to exhibit contributions to the overall aroma, however, it is also reported that a compound with 0.1 < OAV < 1 should also be considered in agreement with the acknowledged theory that sub-threshold compounds may also contribute to wine aroma through additive effects of compounds with similar structure or odor [30]. According to OAV, the volatile compounds that could significantly contribute their flavor notes to the overall aroma of icewines are listed in Table 3. A total of 16 compounds with an OAV above 1 were identified, among them, β-damascenone showed the highest OAV value, followed by ethyl hexanoate. For β-damascenone, BT and SP did not show significant differences, however, in MF and CF it decreased by 20% and 63%, respectively. Furthermore, ethyl hexanoate reduced by 20–80% especially in MF. Other compounds with high OAVs were isoamyl acetate (reduced by 3–33%) and linalool (decreased by 10–20%), but both with OAVs >10. Rose oxide also showed relatively high OAV but was only detected in the control and MF wine samples.

### 2.4. Principal Component Analysis

In order to explore possible differentiation among wine samples clarified by different technologies, principal component analysis (PCA) was applied to the data matrix containing the volatile compounds with OAV > 1 [20,34]. Although the interaction between compounds may be partially lost, these volatiles are the major compounds that contributed to the overall aroma of the wine samples. As principal components 1 and 2 (PC1 and PC2) represented 53.51% and 25.1% of the total variance respectively, approximately 80% of the total variance has been represented (Appendix A). The relationships/correlations between wine samples and compounds could be observed from Figure 2, where the control wine was positioned on the positive side of PC1, while SP and BT wine samples were located on the negative side. Nevertheless, the CF and MF wines were located on the positive and negative sides of PC2, respectively. In addition, rose oxide and eugenol were located on the positive side of PC1 while β-damascenone, citronellol, guaiacol, ethyl hexanoate, ethyl decanoate, octanoic acid, 1-hexnol, isoamyl acetate, ethyl butyrate, and hexanoic acid were located on the negative side (Figure 2 and Appendix A). In addition, these compounds played an important role in aromatic feature which distinguished the control wine from SP and BT wines. As for PC2, ethyl decanoate, ethyl hexanoate, eugenol, and guaiacol were located on the positive side, while octanoic acid, 1-hexnol, isoamyl acetate, ethyl butyrate, hexanoic acid, rose oxide, and geranyl acetone were located on the negative side (Figure 2 and Appendix A) which indicated that CF and MF wines could be separated by these compounds. Crandles et al. also found that most odor-active compounds were positively correlated with icewine cultivar and vintage combinations by PCA analysis [34].

### 2.5. Aroma Series

According to Simonetta et al. [30], wine aroma could be categorized into different series by the aroma wheel. Volatile compounds with OAV values higher than 0.1 were used to evaluate each aroma series in wine samples. In this experiment, icewine aroma was found to consist of floral, fruity, fatty, earthy, spicy, vegetative, and pungent flavor. The floral and fruity aromas were the dominant aromatic categories of ‘Italian Riesling’ icewine, which were mainly contributed by some terpenes (including rose oxide, linalool, citronellol, β-damascenone, and geranyl acetone) or esters (including ethyl butyrate, isoamyl acetate, ethyl hexanoate, ethyl octanoate, and ethyl decanoate compounds) (Table 3). Another study also showed that the compounds with the highest odor activity for icewines were β-damascenone, 1-octen-3-ol, ethyl octanoate, cis-rose oxide and ethyl hexanoate [35], which is similar to the results of this study. Although all tested wine samples exhibited a similar aroma series order, the content of aroma series was different which indicated that clarification treatments played an important role in altering the aromatic quality of the tested wines.

As can be seen from Figure 3, the floral was mainly influenced by mechanical clarification, where MF showed the greatest decrease followed by CF treatment. There were no observed differences between BT and SP fining wines. Fruity was the second largest category of icewine samples, in which MF treatment showed the largest decrease, followed by BT fining, while there was no significant difference between SP and CF wine samples. Fatty, earthy, spicy, vegetative, and pungent flavors could also improve wine complexity. These flavors may be contributed by fatty acids, 1-Octen-3-ol, volatile phenols, C6, and alcohol compounds, however, high levels of these flavors could also negatively influence wine aroma features [33,35,36]. A significant difference in these flavors was found between the control and clarified wines with the exception of earthy (not found in SP wine). MF wine showed the greatest decrease in these flavors, while CF treatment showed the most fatty, earthy, spicy and pungent flavors. Another interesting phenomenon is that clarification treatments showed the least influence in vegetative flavor, where only MF treatment showed a significant difference, but the mechanism still remains to be further studied.

### 2.6. Sensory Evaluation

The icewines treated with different clarification treatments was assessed by sensory analysis to evaluate whether there were any perceivable differences among clarification treatments. As can be seen from Figure 4, clarification treatments could increase limpidity and decrease the color intensity of wine samples. Some researchers suggested that phenol compounds could be absorbed by fining or filtration, thus influence the color intensity of wine samples [17,37], which was in agreement with this study. For balance and persistence, there was no observed differences in the wine samples. Among all treated wines, BT showed the most significant influence on acidity, sweetness and complicacy. Moreover, the aroma intensity, fruity and floral for BT and MF wines were lower than other treatments, indicated that these treatments may have a negative effect on the aroma quality of the wine samples. However, there was no observed difference between BT and MF (Figure 4). In contrast, SP and CF treatments showed little impacts on wine aroma quality (the influence was not significant) which could be alternative method in icewine clarification.

## 3. Materials and Methods

### 3.1. Icewine Samples

The Italian Riesling vines were planted in Gansu Qilian Winery Co., Ltd, Gaotai, Gansu Province, China (latitude 39°18′ N, altitude 99°40′ E). The vines were 22 years old and with a plant density of 1 m × 3 m and the yield was around 4500 kg/ha. Italian Riesling grapes were harvested in November 3–6, 2016 after the first snow and the temperature reduced to −7~−8 °C during the night. The grapes were picked up in the morning when they were frozen and transported to the winery immediately. After bunch sorting, the stems were removed and the grape berries were pressed in a pneumatic press (PADOVAN, Conegliano, Italy). The pressing was done in 3 stages. At the first stage, the pressure was raised to 400 mbar (low pressure) in 10 min and kept for 3 min; at the second stage, the pressure was raised to 800 mbar (middle pressure) in 10 min and kept for another 3 min; finally, the pressure was raised to 1200 mbar (high pressure) and kept for 4 min. Eighty milligram per Liter (80 mg/L) of sulfur dioxide and 35 mg/L of pectinolytic enzyme (Expresser, DSM, Cassano Spinola, Italy) were added during pressing. In this experiment, the free run must and the first pressed must were collected and transferred to a 20 m^3^ stainless steel tank and allowed to settle for 14 h before clarification. Then clean must was obtained by removal of particles and cooled at 0 °C in an insulated tank after which 750 mg/L of bentonite was added and mixed thoroughly. After 8 days of settling, the clean must was further separated/clarified. The chemical variables of the final must were as follows: soluble solids 37.6 °Brix, total sugar content 385.12 g/L, titratable acidity 8.72 g/L, and pH 3.5. A commercial *Saccharomyces cerevisiae* strain (Aroma White, Enartis, San Martino Trecate NO, Italy) was inoculated to induce alcoholic fermentation with a 30 g/hL doses. The fermentation temperature was controlled at 10–12 °C. When the alcohol degree reached 11%/vol, 70 mg/L of sulfur dioxide was added, and the wine was cooled at a temperature of 0 °C to stop alcoholic fermentation. After natural settling, the dead yeast cells were separated from the wine and 40 L of clarified wine was transported to the laboratory and kept at 0–4 °C.

### 3.2. Chemicals and Standards

The standards of volatile compounds were purchased from Sigma-Aldrich (St. Louis, MO, USA), including isobutanol, 3-methyl-1-butanol, *cis*-3-hexen-1-ol, 1-hexanol, heptanol, octanol, phenethyl alcohol, 2,6-nonadien-1-ol, dodecanol, ethyl acetate, isoamyl acetate, ethyl lactate, ethyl hexanoate, ethyl octanoate, diethyl succinate, phenethyl acetate, ethyl decanoate, butanoic acid, hexanoic acid, octanoic acid, dodecanoic acid, octanal, (2e,6z)-2,6-nonadienal, benzaldehyde, furfural, citronellol, linalool, rose oxide, geraniol, nerol, β-damascenone, β-ionone, nerolidol, hexanal, guaiacol, 4-Ethylphenol, and 2-octanol which were used as the internal standard.

Folin–Ciocalteu reagent and sodium chloride were purchased from Beijing Chemical Works (Beijing, China). Potassium hydrogen tartrate were purchased at Kemiou Chemical Reagent Co. (Tianjin, China). Bovine serum albumin was obtained from Asahi Kasei Corporation (Tokyo, Japan). Deionized water (<18 MW resistance) was purified by using a Milli-Q purification system (Molecular, Chongqing, China). Bentonite, soybean protein and potassium metabisulfite were purchased from Lallemand Company (Lallemand, Toulouse, France).

### 3.3. Vinifications and Samples

The different clarification processes were applied as follows: (The diagram of experiment can be found in Appendix A)

(i)8 L of wine without any treatment was kept at 0–4 °C and employed as control © wine;(ii)8 L of wine was clarified with bentonite (BT) and another 8 L of wine was treated with soybean protein (SP) respectively. Different amounts of fining agents were used in preliminary experiment and the stability of wine was tested by heat test. Finally, 1000 mg/L of bentonite and 500 mg/L of soybean protein were used, and the wine was recognized as protein stable. In the fining process, the wine was thoroughly mixed with fining agent solution and held at 0–4 °C for 14 days until the sediments were separated;(iii)Another 8 L of wine was subjected to a plate filter with GS-100 cellulose filter sheets (FLOM-men-0015, Fumei, Xiamen, China) and with a membrane diameter of 0.2 µm. When the filtration was finished, the wine was kept at 0–4 °C in a refrigerator;(iv)Another 8 L of wine was clarified by using a high-speed refrigerated centrifuge (H2050R, Xiangqi, Changsha, China) at 8000 r/min with controlled temperature at 5 °C. The sediments were separated, and the treated wine was kept at 0–4 °C for parameters analysis.

In the clarification processes, a total of 40 L of wine samples were distributed into 2.5 L brown glass tanks, each treatment was performed in triplicate. Samples for analysis were taken from control w©(C), wines clarified with bentonite (BT) and soybean protein (SP), as well as wines submitted to membrane filtration (MF) and centrifugation (CF). Wine samples from each treatment were kept at 0–4 °C in a refrigerator and physical-chemical parameters were measured immediately when the clarification was finished.

### 3.4. Oenological Parameters Analysis

Physico-chemical parameters of the wine samples such as the content of total sugar, total acidity, and volatile acid were measured according to the analytical methods of wine and fruit wine (National Standard of the People’s Republic of China) [38]. The total sugar was measured by Fehling regent titration method whiles the total acidity and volatile acid were measured by acid–alkali titration method, and phenolphthalein was used as the indicator. The transmittance was measured by a Genesis 10S ultraviolet-visible spectrophotometer (Thermo Fisher Scientific, Waltham, MA, USA) at 680 nm. The color intensity and total phenol content were measured according to the Compendium of International Methods of Wine and Must Analysis methods by measuring absorbance at 420 nm and 750 nm (10 mm cell) respectively using the UV-VIS spectrophotometer [39]. The content of protein was measured by Bradford assay [40].

### 3.5. Volatile Analysis

Volatile compounds extraction and analysis were proposed by Duan et al [20]. Five milliliters of icewine, 1.5 g sodium chloride (NaCl), and ten microliter (10 µL) of 2-octanol (internal standard, with the concertation of 88.2 mg/L) were loaded together in a fifteen milliliter vial, and a septum made by polytetrafluoroethylene–silicon was immediately capped. The vial was equilibrated at 40 °C under agitation condition for 30 min, and the volatile components in headspace were adsorbed by a DVB/carboxen/PDMS fiber (50/30 µm, fiber length 1 cm, Supelco, Bellefonte, PA, USA) coupled with a manual holder at above condition [20]. A gas chromatography and a mass spectrometer system (TRACE 1310- ISQ, Thermo Fisher Scientific, San Jose, CA, USA) coupled with a DB-WAX column (60 m × 0.25 mm × 0.25 µm, Agilent Technology, Santa Clara, CA, USA) were used for volatile analysis. The injection port of GC was 250 °C and the volatile compounds were released from the fiber in ten minutes with splitless mode. Helium (99.9% purity) was used as the carrier gas and the flow rate was 1.0 mL/min. Oven temperature was programed as: isothermal at 40 °C for 5 min, heating at 4 °C /min until 200 °C, and kept for 20 min. Mass spectra conditions: electron impact mode (EI) with a 70 eV ionization energy, the ion source and transfer line temperature were set at 250 °C and 200 °C, respectively; mass range was 50–350 and operated with full scan mode (3 scans/s). Volatile identification was confirmed by comparing GC retention time and mass spectra with pure standards. Volatiles without standards were identified by comparing the mass spectra with NIST 11 library (*p* > 90%). The retention indices calculated with C7–C24 n-alkane series (Supelco, Bellefonte, PA, USA) were also compared to the NIST standard reference database. Volatile concentration with standards (reported in Section 3.2) which were calculated by standard curve. Other compounds concentrations were calculated as µg/L of internal standard 2-octanol.

### 3.6. Odour Activity Values and Aroma Series

The odor activity value (OAV) was used to indicate the contribution of different volatile compounds to the overall aroma in the wine. Generally, it is defined as the ratio of concentration over its perception threshold of a volatile compound [30]. According to Simonetta et al., the overall aroma in wine can be categorized into thirteen aroma series including chemical, pungent, oxidized, microbiologic, floral, spicy, fruity, vegetative, nutty, caramelized, woody, earthy and fatty [30]. Aroma series was calculated by summing the OAVs of volatile compounds (OAV > 0.1) in different aromatic categories.

### 3.7. Sensory Analysis

The icewines were evaluated by a panel of 8 judges, comprising members of the staff in viticulture and enology department (age range from 29 to 57 years old, 6 men and 2 women), certificated in wine tasting. The tasting was conducted from 10:00 a.m. to 12:00 p.m. in a standard sensory analysis laboratory (wine tasting room of the College of Food Science and Engineering, Gansu Agricultural University, Lanzhou City, Gansu Province, China) with an individual booth. The samples were served in a random order according to standardized procedures. Twenty milliliters (20 mL) of icewine samples were served into an odor-free tasting glass and the serving temperature was set at 8 ± 2 °C. Prior to the sensory analysis, ten attributes were selected: visual (limpidity and color intensity), aroma (aroma intensity, complicacy, fruity, and floral) and taste (sweetness, acidity, persistence, and balance). The attributes were quantified using a ten-point intensity scale. Low values were “attributed not perceptible” and on the contrast, high values were “attributed strongly perceptible”. The final score was calculated for each wine as the sum of an average score of visual, aroma, and taste attributes [41].

### 3.8. Statistical Analysis

Data were analyzed by using SPSS 19.0 (IBM SPSS, Inc., Chicago, IL, USA). Significance was judged at *p* < 0.05 by analysis of variance (ANOVA) and followed by Duncan’s range test. Furthermore, a principal component analysis (PCA) was carried out on the detected volatile compounds with OAV > 1 by MetaboAnalyst 4.0 after normalization (normalization by median) of data.

## 4. Conclusions

All treatments improved the limpidity of wines. Microfiltration and centrifugation achieved better results, while bentonite and soy protein fining showed higher color intensity. Bentonite fining showed the most significant decrease in total sugar and total acidity content while membrane filtration showed the biggest influence in total phenol content.

Volatile analysis indicated that clarification decreased the concentration of aroma compounds in wines. Among the various clarification treatments, MF treatment showed the most significant influence while SP fining showed much lower impact. Odor activity values evaluation demonstrated that β-damascenone, ethyl hexanoate, rose oxide, isoamyl acetate and linalool presented higher odor activity (>10) being active compounds in icewines. The floral and fruity aromas were found to be the dominant aromatic series of ‘Italian Riesling’ ice-wine, followed by fatty, earthy, spicy, vegetative and pungent flavors. These series decreased significantly in clarified wines where membrane filtration showed the most significant influence in floral, fruity, fatty and spicy aromas while earthy were mainly influenced by soybean protein fining.

Sensory evaluation demonstrated that clarification could improve limpidity and decrease the color intensity of wine samples. The aroma and taste properties of the wine samples were more influenced by bentonite fining, while membrane filtration mainly influenced color and aroma. SP and CF treatments achieved better sensory quality. These results suggested that clarification possessed significant influence on wine composition, thus soybean protein and centrifugation could be an alternative method in icewine clarification to minimize possible negative effects on wine quality. Moreover, clarification is a continuous and fast technique that do not produce adjuvant wastes as the other techniques.

## Figures and Tables

**Figure 1 molecules-25-02657-f001:**
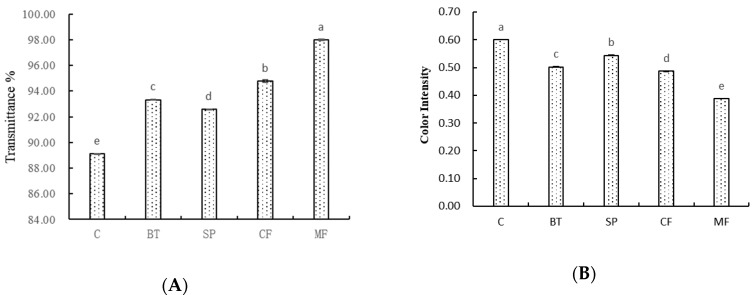
Change of (**A**) light transmittance and (**B**) color intensity in wine samples with different treatments. Different letters represent significant differences at a significant level of 0.05.

**Figure 2 molecules-25-02657-f002:**
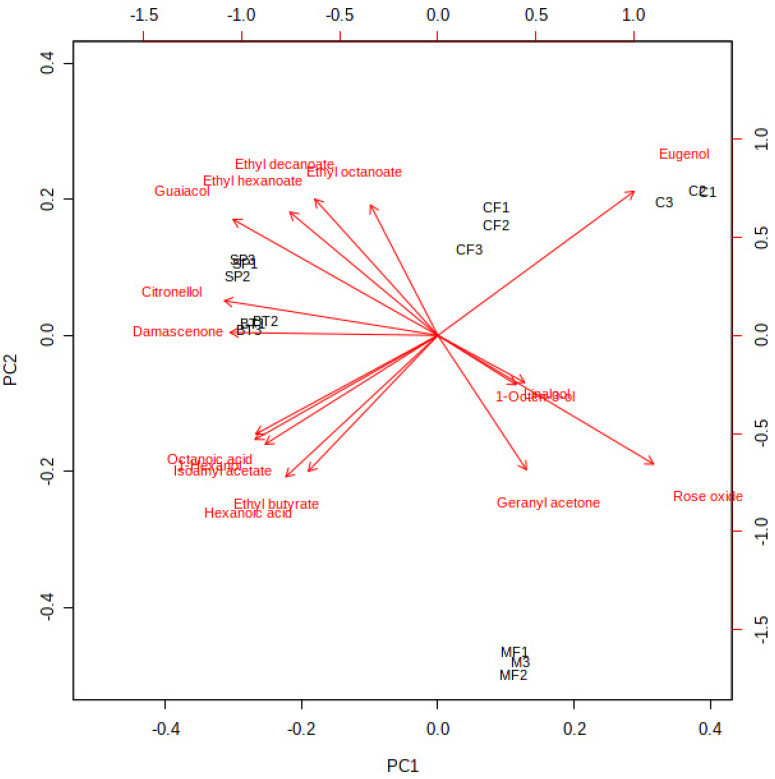
Principal component analysis of volatile compounds (OAV > 1) in wine samples with different clarification treatments.

**Figure 3 molecules-25-02657-f003:**
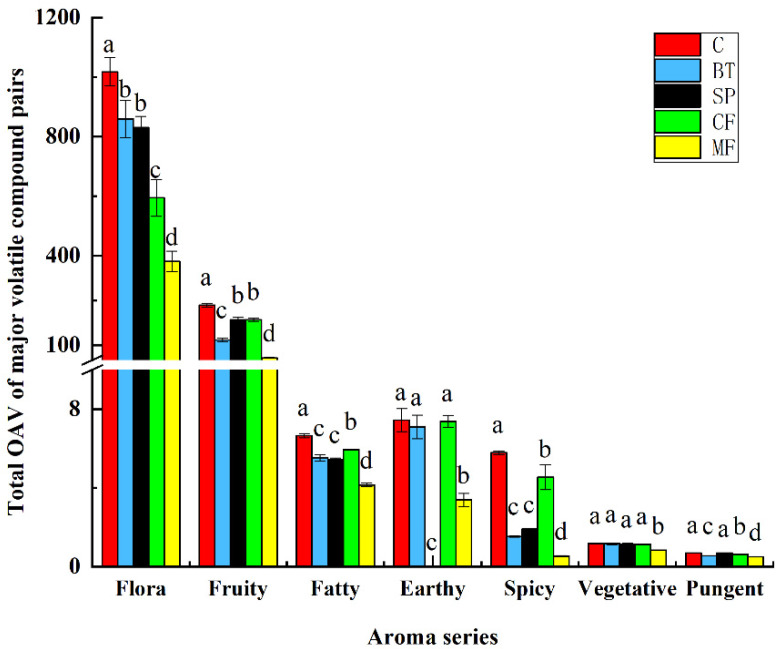
Total OAV of the aroma series after different clarification treatments. Data are means ± SD (*n* = 3). Different letters represent significant differences at a significant level of 0.05.

**Figure 4 molecules-25-02657-f004:**
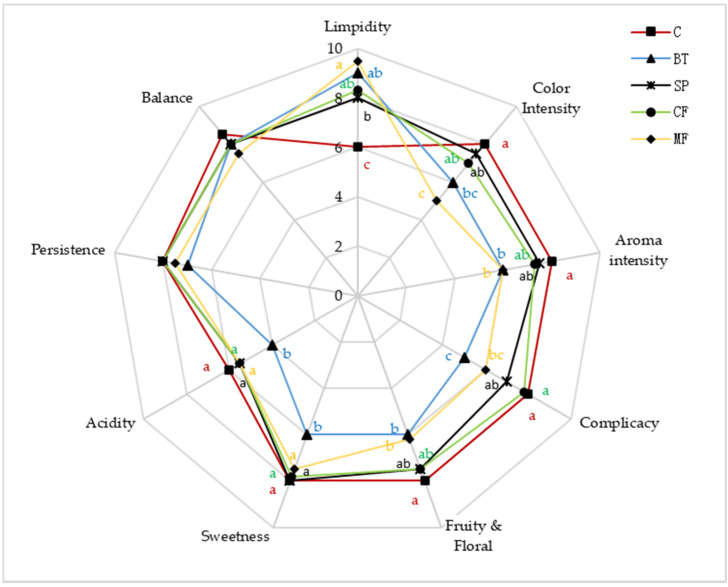
Radar map of sensory analysis. Different letters represent significant differences at a significant level of 0.05.

**Table 1 molecules-25-02657-t001:** Basic physicochemical indexes of wine samples processed under different clarification process.

Treatment	Total Sugar (g/L)	Total Acidity (g/L)	Volatile Acidity (g/L)	Protein (g/L)	Total Phenol (g/L)
C	190.00 ± 1.00 ^a^	9.74 ± 0.01 ^a^	0.51 ± 0.02 ^a^	0.115 ± 0.001 ^a^	0.477 ± 0.000 ^a^
BT	171.33 ± 3.21 ^b^	9.41 ± 0.05 ^d^	0.45 ± 0.02 ^c^	0.114 ± 0.001 ^ab^	0.454 ± 0.011 ^b^
SP	188.67 ± 0.58 ^a^	9.50 ± 0.03 ^bc^	0.50 ± 0.01 ^ab^	0.116 ± 0.000 ^a^	0.441 ± 0.006 ^b^
CF	189.33 ± 1.53 ^a^	9.45 ± 0.05 ^cd^	0.48 ± 0.01 ^b^	0.113 ± 0.001 ^ab^	0.470 ± 0.006 ^a^
MF	190.00 ± 1.00 ^a^	9.55 ± 0.03 ^b^	0.49 ± 0.01 ^ab^	0.115 ± 0.001 ^a^	0.392 ± 0.011 ^c^

Data are means ± SD (*n* = 3). Different letters represent significant differences at a significant level of 0.05.

**Table 2 molecules-25-02657-t002:** Volatile compounds and concentration (mg/L) with standard deviation (SD) of ‘Italian Riesling’ icewine samples.

Compounds	RI DB-WAX	C mg/L ± SD	BT mg/L ± SD	SP mg/L ± SD	CF mg/L ± SD	MF mg/L ± SD
Esters						
Ethyl acetate	883	4.498 ± 0.065 ^a^	3.453 ± 0.144 ^c^	4.465 ± 0.115 ^a^	4.052 ± 0.116 ^b^	3.141 ± 0.064 ^d^
Isobutyl acetate	1006	0.002 ± 0.000 ^a^	0.001 ± 0.000 ^d^	0.002 ± 0.000 ^b^	0.001 ± 0.000 ^c^	ND
Ethyl butyrate	1044	0.049 ± 0.002 ^a^	0.027 ± 0.001 ^e^	0.042 ± 0.001 ^c^	0.046 ± 0.001 ^b^	0.034 ± 0.002 ^d^
Isoamyl acetate	1120	0.493 ± 0.012 ^a^	0.353 ± 0.006 ^c^	0.463 ± 0.007 ^b^	0.449 ± 0.028 ^b^	0.330 ± 0.007 ^c^
Ethyl hexanoate	1232	2.697 ± 0.085 ^a^	1.356 ± 0.074 ^c^	2.142 ± 0.120 ^b^	2.158 ± 0.061 ^b^	0.568 ± 0.023 ^d^
Ethyl (*E*)-hex-2-enoate	1245	0.040 ± 0.001 ^a^	0.021 ± 0.001 ^c^	0.038 ± 0.001 ^b^	0.036 ± 0.004 ^b^	ND
Ethyl heptanoate	1332	0.045 ± 0.001 ^a^	0.016 ± 0.001 ^d^	0.034 ± 0.001 ^b^	0.029 ± 0.002 ^c^	0.003 ± 0.001 ^e^
Ethyl octanoate	1434	2.870 ± 0.080 ^a^	0.768 ± 0.022 ^c^	1.706 ± 0.086 ^b^	1.790 ± 0.140 ^b^	0.574 ± 0.027 ^d^
Ethyl nonanoate	1530	0.005 ± 0.000 ^a^	ND	0.002 ± 0.000 ^b^	ND	ND
Ethyl decanoate	1638	1.737 ± 0.026 ^a^	0.758 ± 0.013 ^d^	1.249 ± 0.039 ^b^	1.023 ± 0.05 ^c^	0.27 ± 0.034 ^e^
Ethyl benzoate	1644	0.028 ± 0.001 ^a^	0.021 ± 0.001 ^c^	0.028 ± 0.002 ^ab^	0.027 ± 0.001 ^b^	ND
Diethyl succinate	1687	0.773 ± 0.009 ^a^	0.653 ± 0.023 ^c^	0.726 ± 0.017 ^b^	0.717 ± 0.030 ^b^	0.695 ± 0.036 ^bc^
Ethyl phenylacetate	1776	0.019 ± 0.002 ^a^	0.014 ± 0.001 ^b^	0.015 ± 0.001 ^b^	0.018 ± 0.001 ^a^	ND
Phenethyl acetate	1825	0.092 ± 0.001 ^a^	0.090 ± 0.003 ^a^	0.089 ± 0.004 ^a^	0.088 ± 0.003 ^a^	0.034 ± 0.001 ^b^
Ethyl dodecanoate	1847	0.727 ± 0.026 ^a^	0.444 ± 0.023 ^c^	0.655 ± 0.033 ^b^	0.377 ± 0.017 ^d^	0.075 ± 0.007 ^e^
Ethyl 3-phenylpropionate	1914	0.010 ± 0.001 ^a^	0.007 ± 0.001 ^b^	0.010 ± 0.001 ^a^	0.010 ± 0.002 ^a^	ND
Ethyl myristate	2043	0.216 ± 0.018 ^a^	0.126 ± 0.015 ^b^	0.205 ± 0.007 ^a^	0.142 ± 0.012 ^b^	0.032 ± 0.002 ^c^
Diethyl malate	2060	0.029 ± 0.001 ^a^	ND	0.028 ± 0.001 ^ab^	0.026 ± 0.003 ^bc^	0.024 ± 0.001 ^c^
Ethyl hexadecanoate	2243	0.134 ± 0.005 ^a^	0.13 ± 0.009 ^ab^	0.123 ± 0.001 ^b^	0.109 ± 0.006 ^c^	0.040 ± 0.001 ^d^
δ-Dodecalactone	2438	0.044 ± 0.001 ^a^	0.039 ± 0.002 ^bc^	0.037 ± 0.001 ^c^	0.042 ± 0.003 ^ab^	ND
	Subtotal	14.508 ± 0.162 ^a^	8.274 ± 0.176 ^d^	12.058 ± 0.235 ^b^	11.141 ± 0.420 ^c^	5.821 ± 0.162 ^e^
Alcohols						
Isobutanol	1077	0.109 ± 0.001 ^a^	0.091 ± 0.002 ^b^	0.109 ± 0.003 ^a^	0.104 ± 0.004 ^a^	0.084 ± 0.005 ^c^
n-Butanol	1150	0.018 ± 0.001 ^a^	0.010 ± 0.001 ^bc^	0.011 ± 0.001 ^b^	0.009 ± 0.001 ^c^	0.010 ± 0.001 ^bc^
1-Pentanol	1255	5.989 ± 0.008 ^a^	5.558 ± 0.181 ^c^	5.980 ± 0.105 ^a^	5.778 ± 0.023 ^b^	4.910 ± 0.124 ^d^
3-Methyl-1-pentanol	1318	0.006 ± 0.001 ^a^	0.005 ± 0.000 ^b^	0.005 ± 0.000 ^b^	0.006 ± 0.000 ^a^	0.004 ± 0.000 ^c^
1-Hexanol	1353	1.098 ± 0.011 ^a^	1.117 ± 0.038 ^a^	1.081 ± 0.035 ^a^	1.064 ± 0.033 ^a^	0.819 ± 0.018 ^b^
*cis*-3-Hexen-1-ol	1366	0.067 ± 0.001 ^a^	0.059 ± 0.002 ^b^	0.066 ± 0.001 ^a^	0.065 ± 0.005 ^a^	0.035 ± 0.002 ^c^
1-Octen-3-ol	1447	0.147 ± 0.012 ^a^	0.141 ± 0.012 ^a^	ND	0.146 ± 0.006 ^a^	0.067 ± 0.007 ^b^
Heptanol	1449	0.064 ± 0.001 ^a^	0.053 ± 0.003 ^c^	ND	0.064 ± 0.002 ^a^	0.058 ± 0.003 ^b^
1-Octanol	1554	0.052 ± 0.001 ^a^	0.052 ± 0.004 ^a^	0.047 ± 0.001 ^a^	0.050 ± 0.006 ^a^	0.045 ± 0.003 ^a^
2,3-Butanediol	1556	0.047 ± 0.002 ^a^	0.042 ± 0.002 ^b^	ND	0.046 ± 0.004 ^a^	ND
Nonanol	1666	0.019 ± 0.001 ^a^	0.014 ± 0.001 ^b^	0.015 ± 0.001 ^b^	0.019 ± 0.001 ^a^	ND
Decanol	1769	0.022 ± 0.001 ^a^	ND	0.021 ± 0.001 ^a^	0.021 ± 0.001 ^a^	0.020 ± 0.001 ^b^
Phenethyl alcohol	1912	2.740 ± 0.010 ^a^	2.719 ± 0.099 ^a^	2.685 ± 0.111 ^a^	2.569 ± 0.219 ^a^	2.312 ± 0.085 ^b^
Dodecanol	1970	0.016 ± 0.001 ^a^	0.016 ± 0.001 ^a^	0.014 ± 0.001 ^b^	0.016 ± 0.001 ^a^	0.015 ± 0.001 ^a^
	Subtotal	10.393 ± 0.031 ^a^	9.876 ± 0.312 ^b^	10.033 ± 0.211 ^ab^	9.956 ± 0.235 ^b^	8.379 ± 0.240 ^c^
Acids						
Acetic acid∗	1452	0.621 ± 0.018 ^a^	0.593 ± 0.013 ^a^	0.606 ± 0.016 ^a^	0.603 ± 0.078 ^a^	0.451 ± 0.022 ^b^
Isobutanoic acid	1581	0.012 ± 0.001 ^a^	ND	0.012 ± 0.001 ^a^	0.011 ± 0.001 ^ab^	0.010 ± 0.000 ^b^
2-Methyl butanoic acid	1655	0.056 ± 0.001 ^a^	0.054 ± 0.001 ^ab^	0.052 ± 0.001 ^b^	0.056 ± 0.002 ^a^	0.052 ± 0.002 ^b^
Hexanoic acid	1851	0.652 ± 0.024 ^a^	0.617 ± 0.024 ^abc^	0.598 ± 0.029 ^bc^	0.640 ± 0.024 ^ab^	0.591 ± 0.017 ^c^
Heptoic acid	1960	0.029 ± 0.002 ^a^	0.027 ± 0.001 ^bc^	0.026 ± 0.001 ^bc^	0.028 ± 0.002 ^ab^	0.025 ± 0.001 ^c^
Octanoic acid	2050	1.237 ± 0.037 ^a^	1.091 ± 0.086 ^b^	1.060 ± 0.052 ^b^	1.122 ± 0.034 ^b^	0.797 ± 0.021 ^c^
Nonanoic acid	2169	0.036 ± 0.001 ^a^	0.016 ± 0.001 ^c^	ND	0.036 ± 0.002 ^a^	0.025 ± 0.001 ^b^
Decanoic acid	2279	0.757 ± 0.011 ^a^	0.596 ± 0.016 ^b^	0.506 ± 0.019 ^c^	0.745 ± 0.013 ^a^	0.109 ± 0.006 ^d^
Dodecanoic acid	2502	0.072 ± 0.001 ^a^	ND	0.035 ± 0.003 ^c^	0.069 ± 0.002 ^a^	0.064 ± 0.001 ^b^
	Subtotal	3.399 ± 0.049 ^a^	2.994 ± 0.107 ^c^	2.860 ± 0.068 ^c^	3.241 ± 0.074 ^b^	2.061 ± 0.054 ^d^
Terpenes						
Rose oxide	1337	0.020 ± 0.001 ^a^	ND	ND	ND	0.005 ± 0.001 ^b^
Linalool	1552	0.501 ± 0.009 ^a^	ND	0.489 ± 0.003 ^ab^	0.402 ± 0.013 ^c^	0.477 ± 0.021 ^b^
α-Terpineol	1680	0.049 ± 0.002 ^a^	0.049 ± 0.001 ^ab^	0.045 ± 0.002 ^ab^	0.045 ± 0.003 ^ab^	0.045 ± 0.003 ^b^
Citronellol	1750	0.401 ± 0.009 ^a^	0.378 ± 0.025 ^a^	0.372 ± 0.015 ^a^	0.387 ± 0.013 ^a^	0.147 ± 0.009 ^b^
Nerol	1806	0.018 ± 0.000 ^a^	0.013 ± 0.001 ^b^	0.013 ± 0.000 ^b^	0.017 ± 0.001 ^a^	0.004 ± 0.001 ^c^
Hydroxycitronellol	1822	0.008 ± 0.001 ^a^	ND	0.008 ± 0.001 ^ab^	0.007 ± 0.001 ^bc^	0.007 ± 0.001 ^c^
β-Damascenone	1831	0.125 ± 0.006 ^a^	0.119 ± 0.009 ^a^	0.112 ± 0.005 ^a^	0.080 ± 0.009 ^b^	0.046 ± 0.004 ^c^
Geranyl acetone	1865	0.100 ± 0.002 ^a^	0.018 ± 0.000 ^c^	0.038 ± 0.007 ^b^	0.098 ± 0.019 ^a^	0.091 ± 0.002 ^a^
Geranic acid	2334	0.007 ± 0.001 ^a^	0.003 ± 0.000 ^c^	0.006 ± 0.001 ^b^	0.006 ± 0.001 ^b^	ND
	Subtotal	1.227 ± 0.022 ^a^	0.582 ± 0.023 ^d^	1.083 ± 0.022 ^b^	1.041 ± 0.038 ^b^	0.823 ± 0.022 ^c^
carbonyls						
6-Methyl-5-hepten-2-one	1341	0.023 ± 0.001 ^a^	ND	ND	0.021 ± 0.001 ^b^	0.021 ± 0.001 ^b^
Oct-2-enal	1434	0.025 ± 0.001 ^b^	0.025 ± 0.001 ^b^	0.027 ± 0.001 ^a^	0.017 ± 0.001 ^c^	ND
Furfural	1468	0.031 ± 0.001 ^ab^	0.033 ± 0.001 ^a^	0.029 ± 0.001 ^b^	0.008 ± 0.002 ^c^	ND
	Subtotal	0.078 ± 0.001 ^a^	0.058 ± 0.001 ^b^	0.056 ± 0.001 ^b^	0.046 ± 0.002 ^c^	0.021 ± 0.001 ^d^
Volatile phenols						
Eugenol	2141	0.022 ± 0.001 ^a^	ND	ND	0.015 ± 0.004 ^b^	ND
Guaiacol	2203	0.043 ± 0.001 ^a^	0.031 ± 0.001 ^d^	0.038 ± 0.002 ^c^	0.041 ± 0.001 ^b^	0.011 ± 0.001 ^e^
	Subtotal	0.065 ± 0.001 ^a^	0.031 ± 0.001 ^d^	0.038 ± 0.001 ^c^	0.056 ± 0.004 ^b^	0.011 ± 0.001 ^e^

Data are means ± SD (*n* = 3). Retention indices (RI) were reported in the NIST standard reference database. Different letters represent significant differences at a significant level of 0.05. “ND” means that the aroma component is not detected or in trace amount. * The concentration of Acetic acid was presented with g/L.

**Table 3 molecules-25-02657-t003:** Odor activity values (OAV > 0.1) and odor description of the would-be impact odorants of different clarified ice-wine samples.

Compounds	Odor Description	Odor Threshold (µg/L)	Reference	Odor Activity Value					Aroma Classes
				C	BT	SP	CF	MF	
Ethyl acetate	Pineapple, fruity, balsamic	7500	**c**	0.60	0.46	0.60	0.54	0.42	2,7
Isobutyl acetate	Fruity, apple	20	a	0.12	0.07	0.11	0.07	0.00	2
Ethyl butyrate	Floral, fruity	20	a	2.47	1.34	2.08	2.30	1.72	1,2
Isoamyl acetate	Banana	30	a	16.45	11.76	15.44	14.97	11.01	2
Ethyl hexanoate	Fruity, green apple	14	a	192.62	96.85	153.02	154.18	40.59	2
Ethyl heptanoate	Pineapple, fruity	220	c	0.21	<0.1	0.16	0.13	<0.1	2
Ethyl octanoate	Ripe fruits, pear, sweety	240	a	11.96	3.20	7.11	7.46	2.39	2
Ethyl decanoate	Pleasant fruity	200	b	8.68	3.79	6.25	5.11	1.35	2
Diethyl succinate	Fruity, cheese	6000	c	0.13	0.11	0.12	0.12	0.12	2,3
Ethyl phenylacetate	Floral, honey	73	c	0.25	0.19	0.20	0.24	0.00	1
Phenethyl acetate	Floral	250	a	0.37	0.36	0.36	0.35	0.14	1
Ethyl dodecanoate	Oily, fatty, fruity	1500	b	0.48	0.30	0.44	0.25	0.05	2,3
δ-Dodecalactone	Coconut fruity	500	c	0.09	0.08	0.07	0.08	0.00	2
Ethyl myristate	Mild waxy, soapy	500	d	0.43	0.25	0.41	0.28	0.06	3
Ethyl hexadecanoate	Mild waxy	1000	d	0.13	0.13	0.12	0.11	0.04	3
1-Pentanol	Balsamic, bitter almond	64,000	b	0.09	0.09	0.09	0.09	0.08	7
1-Hexanol	Herbaceous, grass, woody	1100	b	1.00	1.02	0.98	0.97	0.74	6
*cis*-3-Hexen-1-ol	Plant, fruity, aromatic	400	a	0.17	0.15	0.16	0.16	0.09	6,2
1-Octen-3-ol	Mushroom	20	d	7.37	7.05	ND	7.30	3.34	4
Heptanol	Oily	200	b	0.32	0.27	ND	0.32	0.29	3
1-Octanol	Jasmine, lemon	800	b	0.06	0.06	0.06	0.06	0.06	3,1
Decanol	Waxy, fatty	400	b	0.05	0.00	0.05	0.05	0.05	3
Phenethyl alcohol	Flowery, rose, Honey	10,000	a	0.27	0.27	0.27	0.26	0.23	1
2-Methyl butanoic acid	Fatty, rancid, cheesy	250	a	0.22	0.22	0.21	0.22	0.21	3
Hexanoic acid	Cheese, fatty	420	b	1.55	1.47	1.42	1.52	1.41	3
Heptoic acid	Fatty	300	d	0.10	0.09	0.09	0.09	0.08	3
Octanoic acid	Rancid, cheese, fattyacid	500	b	2.47	2.18	2.12	2.24	1.59	3
Nonanoic acid	Fatty acid, dry, woody	1400	a	0.07	0.03	0.00	0.07	0.05	3
Decanoic acid	Fatty acid, dry, woody	1400	a	0.54	0.43	0.36	0.53	0.08	3
Dodecanoic acid	Laurel oil	1000	c	0.07	0.00	0.03	0.07	0.06	3
Rose oxide	Lychee	0.2	b	97.89	ND	ND	ND	26.41	1
Linalool	Flowery, Muscat	25	b	20.03	ND	20.35	16.07	19.08	1
α-Terpineol	Floral	250	a	0.19	0.20	0.18	0.18	0.18	1
Citronellol	Rose	100	b	4.01	3.79	3.52	3.87	1.47	1
β-Damascenone	Rose, sweet, flowers	0.14	b	889.98	852.97	802.89	569.67	330.58	1
Geranyl acetone	Floral	60	c	1.67	0.30	0.63	1.63	1.52	1
Geranic acid	Citric, geranium	20	b	0.33	0.17	0.29	0.30	0.00	1
Eugenol	Clove	6	c	3.64	ND	ND	2.50	ND	5
Guaiacol	Clove, curry	20	d	2.14	1.53	1.90	2.05	0.54	5

Data are mean values. ND means that the aroma component is not detected, or it is in trace amount. Odor description (ODE) and odor threshold (OTH) as reported in the literature references a-[30]; b-[31]; c-[32]; d-[33]. Odor activity value (OAV): defined as the ratio between odor concentration and OTH. Aroma classes: each compound was attributed to one or more classes depending on odor descriptors: 1, floral; 2, fruity; 3, fatty; 4, earthy; 5, spicy; 6, vegetative; 7, pungent.

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
