# Peer review of "Effect of Different Clarification Treatments on the Volatile Composition and Aromatic Attributes of ‘Italian Riesling’ Icewine"

_molecules, 2020, doi:10.3390/molecules25112657_

Round 1

Reviewer 1 Report

This study is conducted to evaluate the influence of different clarification treatments (BT, SP, MF, CF) to volatile composition and aromatic attributes of Italian ice wine. It has detected the clarity, physicochemical indexes, volatile compounds and aromatic attributes of treated wines, and made a comparison of these methods due to the analysis of oenological parameters, volatile compounds, odor activity values, aroma series and sensory analysis. The idea of the experiment is very clear, but the actual value of the research is not clear, and there are still some small problems in the experiment process, as follows.

  1. Abstract, I don’t know the importance of the study, and could you tell me which one is better, in fact, the conclusion is a little bit poor, please revise it .
  2. Line 30-31, this sentence is unclear, please rewrite it.
  3. Introduction, please elaborate more on the expected results and significance of the experiment and why author choose Italian ice wine as the experimental sample.
  4. Line 70, change “analyses” to “analysis”.
  5. Line 101-102, why are there no error bars in these two graphs? Please correct.
  6. Line120-121, this sentence is unclear, is it to clarify the interaction between the treatment and aromatic compounds or anything else?
  7. Line211,please explain the meaning of TPI and CI.
  8. Line216-231, this part does not match the theme of 2.2.5,it is recommended to add this part to the introduction
  9. Line 230, the meaning of the statement is unclear, please put a comma in front of “cause” and change it to “because”.
  10. Line 252, the graph of principal component analysis is not very clear, please modify it.
  11. Line 322, there isn’t total sugar content, please add.
  12. Line 327-328, please describe the fermentation process in detail, including yeast used, process conditions, etc.
  13. Line 337-342, I don’t see the introduction of centrifuges and filters, please add.
  14. Line351-352, please make a detailed description of the specific operation process of filtering.
  15. Line360-361, please specify the measurement method.
  16. Conclusion, the conclusion is relatively superficial and it is suggested to rewrite it.

Author Response

Dear  Reviewer:

Thank you for your comments concerning our manuscript entitled “Effect of Different Clarification Treatments on the Volatile Composition and Aromatic Attributes of ‘Italian Riesling’ Icewine”. These comments are all valuable and very helpful for revising and improving our paper, as well as the important guiding significance to our researches. We have studied the comments carefully and have made the necessary corrections which we hope would meet your approval. Revised portions are marked in red in the paper. We hope this revision be acceptable for publication in Molecules.

Sincerely yours,

Teng-Zhen Ma

19/05/2020

Response to Reviewer 1 Comments

Point 1: Abstract, I don’t know the importance of the study, and could you tell me which one is better, in fact, the conclusion is a little bit poor, please revise it.

Response 1: In north China, icewine is an important product for winemakers as Chinese consumers showed great passion in this dessert wine. Clarity is an essential requirements of icewine, but some clarification technologies have confirmed to reduce the aroma and sensory quality of this product. Therefore, how to maintain or improve the aroma and sensory quality in icewine clarification seems very important to winemakers.

Aside the traditional method of clarification, centrifugation or vegetable protein fining could clarify the wine with less influence on the aroma profiles, but these methods are rarely used in icewine production, their influence need to be studied. The aim of this paper is to provide a useful insight and technical support in icewine clarification,

We can conclude that soybean protein and centrifugation are better for wine aroma quality,moreover, the abstract and conclusion have been revised.

Point 2: Line 30-31, this sentence is unclear, please rewrite it.

Response 2: The sentence has been rewritten.

Point 3: Introduction, please elaborate more on the expected results and significance of the experiment and why author choose Italian ice wine as the experimental sample.

Response 3: Generally, Vidal Blanc and Riesling are more widely used in icewine production. But Italian Riesling can also be used because this variety shows high content in sugar, acids, aroma, and flavor compounds, and is the most important variety in the Hexi Corridor Region of China (The vines were 22 years old).

The expected results and significance of the experiment in the introduction have been elaborated.

Point 4: Line 70, change “analyses” to “analysis”.

Response 4: The word “analyses” have been changed to “analysis”.

Point 5: Line 101-102, why are there no error bars in these two graphs? Please correct.

Response 5: Actually, there are error bars in the two graphs, but the deviation is too low to notice the error bars.

Point 6: Line120-121, this sentence is unclear, is it to clarify the interaction between the treatment and aromatic compounds or anything else?

Response 6: This sentence aims to support why clarification influence wine aroma.

Point 7: Line211,please explain the meaning of TPI and CI.

Response 7: The meaning of TPI and CI has been explained. (total phenol index and color intensity)

Point 8: Line216-231, this part does not match the theme of 2.2.5, it is recommended to add this part to the introduction

Response 8: Line 216-231 has been added to the introduction.

Point 9: Line 230, the meaning of the statement is unclear, please put a comma in front of “cause” and change it to “because”.

Response 9: The word “cause” has been changed to “because” and a comma is also added.

Point 10: Line 252, the graph of principal component analysis is not very clear, please modify it.

Response 10: The graph of PCA analysis has been modified to be more clear, but the the position of volatile compounds cannot move in the software.

Point 11: Line 322, there isn’t total sugar content, please add.

Response 11: The sugar content (385.12g/L) of pressed must has been added.

Point 12: Line 327-328, please describe the fermentation process in detail, including yeast used, process conditions, etc.

Response 12: The yeast was a commercial Saccharomyces cerevisiae named Aroma White produced by Enartis company Italia, this message could be found in line 333. Other process conditions have also been added.

Point 13: Line 337-342, I don’t see the introduction of centrifuges and filters, please add.

Response 13: The introduction of centrifuges and filters were presented at line 364 and 367.

Point 14: Line351-352, please make a detailed description of the specific operation process of filtering.

Response 14: A detailed description of the filtering process has been made.

Point 15: Line360-361, please specify the measurement method.

Response 15: The measurement method has been specified.

Point 16: Conclusion, the conclusion is relatively superficial and it is suggested to rewrite it.

Response 16: The conclusion has been rewritten.

Reviewer 2 Report

The topic of the manuscript, the effect of fining treatments on the aromatic profile of wine, is not original, as many papers have been published on the same argument. The only novelty is the application to a “special” wine such as Icewine.

The setting of the experiment is quite poor, with just a comparison among different treatments, without exploring, for example, different dosages of fining agents. The “bad” performance of BT treatment, for example, could be due to the very high dose used for clarification, while a better effect could have been obtained with lower doses.

In some parts, results and discussion seems more Results, without discussion, I suggest to introduce a larger comparison of the results with those obtained by other authors (the literature on the argument is very rich).

Finally, the English level of the Manuscript is very poor, with, in addition, many digit errors. I suggest a thoroughly revision of the Manuscript by a native speaking before to resubmit the paper to this or to another Journal.

For these reasons, in my opinion this paper can be published on Molecules journal only after major revisions.

In addition some comments that can help in improving the Manuscript:

Lines 92-100: why the MF decrease colour in such a way? It would be expected a higher effect by bentonite or SP treatments. More discussion is needed about this results, with the aid of literature data

Table1: how can you explain the reduction in sugars, acidity and volatile acidity upon BT treatment? Could it be due to a  “dilution” effect due to the very high dosage of bentonite?

Surprisingly the protein content remained the same after bentonite, even in the text the authors report that BT reduced protein content). This is an important issue, as the absence of protein removal can indicate an erroneous application of the fining agent, which could influence even the effect on the other parameters.

In addition you measured only the protein content (with Bradford assay, which is not the best method for protein quantification in wine, see Gazzola et al. 2015, AJEV, 66:227), while a very simple way to measure the bentonite efficacy is the heat test followed by a measurement of turbidity induced by heating.

Even on the aroma compound the effect of MF appears surprising, you can try to justify this effect by comparison with analogous experiments reported in other manuscripts.

The effect of bentonite on aroma compounds can be better compared with other results, I suggest a comparison with Vincenzi et al., 2015, JAFC 63: 2314

Author Response

Dear Reviewer:

Thank you for your comments concerning our manuscript entitled “Effect of Different Clarification Treatments on the Volatile Composition and Aromatic Attributes of ‘Italian Riesling’ Icewine”. These comments are all valuable and very helpful for revising and improving our paper, as well as the important guiding significance to our researches. We have studied the comments carefully and have made the necessary corrections which we hope would meet your approval. Revised portions are marked in red in the paper. We hope this revision be acceptable for publication in Molecules.

Sincerely yours,

Teng-Zhen Ma

19/05/2020

Response to Reviewer 2 Comments

Point 1: different dosages of fining agents.

Response 1: Icewine is more difficult to clarify than dry wines because of its high levels of sugar and soluble solids contents. Thus, winemakers usually use high amount of fining agents to ensure the clarity of wines. In this study, we tried different dosages of fining agents in a preliminary experiment but finally choosed 1000 mg/L of Bentonite and 500 mg/L of Soybean Protein, and the wine was recognised as protein stable by heat test.

Point 2: In some parts, results and discussion seems more Results, without discussion

Response 2: More discussion has been introduced.

Point 3: The English level of the Manuscript is poor.

Response 3: The English language has been revised by a native speaker.

Point 4: Lines 92-100: why the MF decrease color in such a way? It would be expected a higher effect by bentonite or SP treatments. More discussion is needed about this results, with the aid of literature data.

Response 4: The membrane filtration influenced color due to the membrane material and diameter used. In this study, the membrane was made by cellulose acetate and the diameter was 0.2µm, thus MF treatment showed the most significant difference in wine color. Other researches also proved bentonite and soybean decreased wine color but not significant compared to PVPP fining.

More discussion was induced in the revised paper.

Point 5: Table1: how can you explain the reduction in sugars, acidity, and volatile acidity upon BT treatment? Could it be due to a “dilution” effect due to the very high dosage of bentonite?

Response 5: The reduction in sugars, acidity, and volatile acidity upon BT treatment may be due to the adsorption mechanism. We can also conclude that the content of residual sugar and total acidity decreased in other research (Lambri M et al. 2012). Considering icewine contains more sugar and acids, there would be a large reduction in these parameters when a very high dosage of bentonite is used.

Point 6: Surprisingly the protein content remained the same after bentonite, even in the text the authors report that BT reduced protein content). This is an important issue, as the absence of protein removal can indicate an erroneous application of the fining agent, which could influence even the effect on the other parameters.

Response 6: Generally, the protein content would reduce after clarification,but in this research, we did not find a significant decrease in protein content. Wine protein could come from grape berry, yeast autolysis, and winemaking procedures. In this study, as the must was clarified by bentonite before fermentation, and the yeast cells were immediately separated after alcoholic fermentation, thus the protein content of experiment wine was quite low and caused no obvious change in the protein content.

Besides, we also detected the protein stability (by following the method described by Pocock et al., 2006) by heat test after clarification treatment, which found that the wine was protein stable (the results did not show in the paper).

Point 7: In addition you measured only the protein content (with Bradford assay, which is not the best method for protein quantification in wine, see Gazzola et al. 2015, AJEV, 66:227), while a very simple way to measure the bentonite efficacy is the heat test followed by a measurement of turbidity induced by heating.

Response 7: The reason to detect the total protein in this experiment is to demonstrate there is no residual soybean protein in treated wines, and as mentioned in response 6, the wine had good protein stability, which indicate Bradford assay could be used. Indeed this traditional assay it is not the best method and was strongly affected by protein type, KDS/BCA assay provides better superiority and will be used in further study.

Point 8: Even on the aroma compound the effect of MF appears surprising, you can try to justify this effect by comparison with analogous experiments reported in other manuscripts.

Response 8: In this research, the membrane filtration influenced wine aroma surprisingly could be due to the filtration method, the membrane material and diameter used. In this study, the traditional and lab-scale dead-end filtration coupled with cellulose acetate membrane with a small diameter in symmetric structure influenced the wine aroma significantly. Nowadays, cross-flow filtration coupled with hollow fibre membranes in asymmetric structure shows higher filtration efficiency and less impact on wine quality, we also discussed this new method in this paper.

Point 9: T. The effect of bentonite on aroma compounds can be better compared with other results, I suggest a comparison with Vincenzi et al., 2015, JAFC 63: 2314

Response 9: The effect of bentonite on aroma compounds has been compared with Vincenzi et al., 2015, JAFC 63: 2314. (see line 154,189 and 214)

Reviewer 3 Report

Dear Authors,

you can find below my comments and suggestions:

Introduction: Sentence ”Among them, bentonite is still the most efficient...reduces both wine quantity and quality.” - This sentence needs supporting references and additional argumentation. The sentence” Thus, plant-based products with lower allergenic...interest.” needs citing references and data to support it. Pay attention to this sentence because it can be strongly linked to the impact of this study. So, I suggest the authors insist on soybean protein advantages (with clear data). Moreover, as the idea is not a new one in the wine research (see Application of soy protein isolate in the fining of red wine - Ficagna et al., 2019), the authors must point out the novelty brought by their findings. 

Results and discussion: Line 127 - ”66 kinds of volatile compounds” - please rephrase this part. There are 66 different volatile compounds. Term ”kind” might be confused with the meaning ”category/class”.

The sentence ”Compared to crude wine...in MF treated wines.” - please refer to the control sample instead of using the term ”crude”. Please rephrase and adjust the argumentation of this part. Not always the decrease in volatile compounds is considered a negative effect. So, the authors must argument which compounds are responsible for this and explain the reason why it is considered a negative effect. Please consider also the ratio of some volatile compounds. 

Fatty acids section - Pay attention to identified acids. Not all acids referred to in this section are fatty acids. 

Generally, there is a lack is discussion of the obtained results with previous data. Not enough explanations are given.

Materials and methods: Please provide more data about the equipment used (pneumatic press, fermentation tank - model, producer, country, capacity). More details must be given regarding the end of alcoholic fermentation: sugar content, alcohol content, sulphur dioxide content. What was the indicator of alcoholic fermentation ending? Authors must prove that no biochemical change was made between experimental stages. Usually, samples are kept at freezing temperature before analytical experiments, otherwise, errors arise. 

Vinifications and samples sections - (i) - please specify the temperature and duration control sample was kept until the analytical experiments were performed. 

(iii) - the same comments as for point  (i). Additionally, please specify the temperature these samples were obtained.

(iv) - please specify the temperature and storage duration.

Lines 355-356 - Sentence is confusing if we look also to the volumes taken in each variant (8/16/8/8). Please explain.

The number of repetitions is not specified.

An experimental diagram including the samples abbreviation will be helpful for the reader. 

Line 396 - For the definition of OAV one citing reference is enough. Please choose one. 

Sensory analysis section - Is not clear in which institution the sensory analysis was performed.  

line 396 - There are citing references missing from the reference list. Please revise the list. 

Spelling and grammar errors were encountered. English proofing is imposed by a native English speaker. 

Author Response

Dear Reviewer:

Thank you for your comments concerning our manuscript entitled “Effect of Different Clarification Treatments on the Volatile Composition and Aromatic Attributes of ‘Italian Riesling’ Icewine”. These comments are all valuable and very helpful for revising and improving our paper, as well as the important guiding significance to our researches. We have studied the comments carefully and have made the necessary corrections which we hope would meet your approval. Revised portions are marked in red in the paper. We hope this revision be acceptable for publication in Molecules.

Sincerely yours,

Teng-Zhen Ma

19/05/2020

Response to Reviewer 3 Comments

Point 1: Introduction: Sentence ”Among them, bentonite is still the most efficient...reduces both wine quantity and quality.” - This sentence needs supporting references and additional argumentation. The sentence” Thus, plant-based products with lower allergenic...interest.” needs citing references and data to support it. Pay attention to this sentence because it can be strongly linked to the impact of this study. So, I suggest the authors insist on soybean protein advantages (with clear data). Moreover, as the idea is not a new one in the wine research (see Application of soy protein isolate in the fining of red wine - Ficagna et al., 2019), the authors must point out the novelty brought by their findings

Response 1: supporting references and additional argumentation has been added in these sentences.

Point 2: Results and discussion: Line 127 - ”66 kinds of volatile compounds” - please rephrase this part. There are 66 different volatile compounds. Term ”kind” might be confused with the meaning ”category/class”.

Response 2: The word ”kind” cannot be used to describe volatile compounds, this sentence has been rewritten as suggested.

Point 3: The sentence ”Compared to crude wine...in MF treated wines.” - please refer to the control sample instead of using the term ”crude”. Please rephrase and adjust the argumentation of this part. Not always the decrease in volatile compounds is considered a negative effect. So, the authors must argument which compounds are responsible for this and explain the reason why it is considered a negative effect. Please consider also the ratio of some volatile compounds.

Response 3: Some papers use the word crude to describe the original sample wines but in this research, as the reviewer suggested, t the word “crude” changed to “control” is a better way to describe the unfined wine, a total of 11 “crude” have been changed to “control”.

Indeed, not always the decrease in volatile compounds is considered a negative effect, the sentence has been rephrased, and further explanation could be found in the revised paper.

Point 4: Fatty acids section - Pay attention to identified acids. Not all acids referred to in this section are fatty acids.

Response 4: As the reviewer mentioned, not all acids are fatty acids, the sentence has been rewritten.

Point 5: Generally, there is a lack in the discussion of the obtained results with previous data. Not enough explanations are given.

Response 5: Discussion of the obtained results with previous data has been added.

Point 6: Materials and methods: Please provide more data about the equipment used (pneumatic press, fermentation tank - model, producer, country, capacity). More details must be given regarding the end of alcoholic fermentation: sugar content, alcohol content, sulphur dioxide content. What was the indicator of alcoholic fermentation ending? Authors must prove that no biochemical change was made between experimental stages. Usually, samples are kept at freezing temperature before analytical experiments, otherwise, errors arise.

Response 6: More data about the equipment have been provided. Indeed, samples are usually kept at freezing temperatures to avoid biochemical change before analytical experiments. In this experiment, the wine was kept at 0℃. The biological stability of the icewine was acceptable because of the relatively high sugar, acid, and sulphur dioxide content, as well as the alcohol degree and storage temperature of sample wines. Besides, when the clarification treatment finished, the analytical experiments proceeded immediately and the sugar content and volatile acidity did not change thus, no biochemical change was made in this severe condition.

Point 7: Vinifications and samples sections - (i) - please specify the temperature and duration control sample was kept until the analytical experiments were performed.

(iii) - the same comments as for point (i). Additionally, please specify the temperature these samples were obtained.

(iv) - please specify the temperature and storage duration.

Response 7: the temperature and storage duration have been added.

Point 8: Lines 355-356 - Sentence is confusing if we look also to the volumes taken in each variant (8/16/8/8). Please explain.

Response 8: Sixteen Litres (16 L) of wine was clarified by fining agents which was then separated into 2 parts of 8 L each. One part (8 L) was treated with bentonite, and another part (8 L) was treated with soybean protein. The confusing sentence has been modified.

Point 9: An experimental diagram including the samples abbreviation will be helpful for the reader.

Response 9: The experimental diagram has been added as supplementary material as figure S3.

Point 10: Line 396 - For the definition of OAV one citing reference is enough. Please choose one.

Response 10: The definition of OAV was cited by reference [29].

Point 11: Sensory analysis section - Is not clear in which institution the sensory analysis was performed.

Response 11: The sensory analysis was performed in sensory room of the college of food science and engineering, Gansu Agricultural University, Lanzhou City, Gansu Province, China. The institution has added in article.

Point 12: line 396 - There are citing references missing from the reference list. Please revise the list.

Response 12: The reference list has been revised and missing citations have been added.

Point 13: Spelling and grammar errors were encountered. English proofing is imposed by a native English speaker.

Response 13: The English language has been revised by a native English speaker.

Round 2

Reviewer 3 Report

Please check the manuscript carrefully as there are still some spelling errors (one example ”3.5. Volitate Analysis”).

You did not detect 57 categories of compounds, you detected 57 volatile compounds. Please correct the sentence (line 136).

Author Response

Dear Reviewer:

Thank you for your comments concerning our manuscript entitled “Effect of Different Clarification Treatments on the Volatile Composition and Aromatic Attributes of ‘Italian Riesling’ Icewine”. These comments are valuable and very helpful for revising and improving our paper. We have made the corrections by “Track Changes” function in Microsoft Word, and we hope this revision would meet your approval.

Sincerely yours,

Teng-Zhen Ma

1/06/2020

 Response  for Reviewers

Point 1: Please check the manuscript carefully as there are still some spelling errors (one example ”3.5. Volitate Analysis”).

Response 1: the manuscript has been carefully checked, spelling and grammar errors were corrected. The "Track Changes" function in Microsoft Word was used to clearly highlighted the revisions.

Point 2: You did not detect 57 categories of compounds, you detected 57 volatile compounds. Please correct the sentence (line 136).

Response 2: This sentence and other similar mistakes has been corrected.